# Fairness constraints can help exact inference in structured prediction

**Kevin Bello**
Department of Computer Science
Purdue Univeristy
West Lafayette, IN 47906, USA
kbellome@purdue.edu

**Jean Honorio**
Department of Computer Science
Purdue Univeristy
West Lafayette, IN 47906, USA
jhonorio@purdue.edu

## Abstract

Many inference problems in structured prediction can be modeled as maximizing a score function on a space of labels, where graphs are a natural representation to decompose the total score into a sum of unary (nodes) and pairwise (edges) scores. Given a generative model with an undirected connected graph $G$ and true vector of binary labels $\overline{y}$, it has been previously shown that when $G$ has good expansion properties, such as complete graphs or $d$-regular expanders, one can exactly recover $\overline{y}$ (with high probability and in polynomial time) from a single noisy observation of each edge and node. We analyze the previously studied generative model by Globerson et al. (2015) under a notion of statistical parity. That is, given a *fair* binary node labeling, we ask the question whether it is possible to recover the fair assignment, with high probability and in polynomial time, from single edge and node observations. We find that, in contrast to the known trade-offs between fairness and model performance, the addition of the fairness constraint *improves* the probability of exact recovery. We effectively explain this phenomenon and empirically show how graphs with poor expansion properties, such as grids, are now capable of achieving exact recovery. Finally, as a byproduct of our analysis, we provide a tighter *minimum-eigenvalue* bound than that which can be derived from Weyl's inequality.

## 1 Introduction

Structured prediction consists of receiving a structured input and producing a combinatorial structure such as trees, clusters, networks, sequences, permutations, among others. From the computational viewpoint, structured prediction is in general considered intractable because of the size of the output space being exponential in the input size. For instance, in image segmentation tasks, the number of admissible segments is exponential in the number of pixels. In this work, we focus on the inference problem, where a common approach is to exploit local features to infer a global structure.

Consider an undirected graph $G = (V, E)$. In graphical models, specifically in Markov random fields, one generally tries to find a solution to the following optimization problem:

$$\max_{\boldsymbol{y} \in \mathcal{M}^{|V|}} \sum_{v \in V, m \in \mathcal{M}} c_v(m) \mathbb{1}[y_v = m] + \sum_{(u,v) \in E, m_1, m_2 \in \mathcal{M}} c_{u,v}(m_1, m_2) \mathbb{1}[y_u = m_1, y_v = m_2], \quad (1)$$

where $\mathcal{M}$ is the set of possible labels, $c_u(m)$ is the cost or potential of assigning label $m$ to node $v$, and $c_{u,v}(m, n)$ is the cost or potential of assigning $m$ and $n$ to the neighbors $u, v$ respectively. This type of inference problem arises in the context of community detection, statistical physics, sociology, among others. Only a few particular cases of problem 1 are known to be solvable in polynomial time. To name a few, (Schraudolph & Kamenetsky 2009) and (Chandrasekaran et al. 2008) showed that

problem 1 can be solved exactly in polynomial time for planar ising models and graphs with low treewidth, respectively.

As the use of machine learning in decision making increases in our society (Kleinberg et al. 2018), researchers have shown interest in developing methods that can mitigate unfair decisions or avoid bias amplification. With the existence of several notions of fairness (Gajane & Pechenizkiy 2017, Verma & Rubin 2018, Barocas & Selbst 2016, Feldman et al. 2015), and some of them being simultaneously incompatible (Kleinberg et al. 2016), the first step is to define the notion of fairness, which is commonly dependent upon the task on hand. For our purposes, we will adapt the notion of statistical parity and apply it to the exact inference problem. Several notions of statistical parity have been studied in prior works (Agarwal et al. 2019, Johnson et al. 2016, Calders et al. 2013), where, in general, statistical parity enforces a predictor to be independent of the protected attribute. In particular, in regression, (Agarwal et al. 2019) relaxed the principle of statistical parity and studied $\varepsilon$-away difference of marginal CDF and conditional CDF on the protected attribute. Finally, unlike the works on supervised learning (Hardt et al. 2016, Luong et al. 2011, Agarwal et al. 2018), the work of (Chierichetti et al. 2017) is among the first to adapt the disparate impact doctrine (related to statistical parity) to unsupervised learning, specifically, to the clustering problem.

We study a generative model similar to the one that has been previously studied in (Globerson et al. 2015, Foster et al. 2018, Bello & Honorio 2019, Abbe et al. 2016), and whose objective follows a similar form of problem 1, with the addition of a fairness constraint. While (Globerson et al. 2015, Foster et al. 2018) studied the regime of approximate inference, (Abbe et al. 2016, Bello & Honorio 2019) studied the scenario of exact inference. The latter authors showed that it suffices to have graphs with high expansion properties to efficiently achieve exact recovery with high probability. However, graphs such as grids remained evasive to exact recovery due to their poor expansion properties.

**Contributions.** We propose a generative model, similar to that of (Globerson et al. 2015), where the true labeling is *fair*, and ask the following question: Will the addition of a fairness constraint in the inference problem have any effect on the probability of exact recovery? In spite of the results on inherent trade-offs of fairness and performance (Zhao & Gordon 2019, Kleinberg et al. 2016), we show that the addition of a fairness constraint, in this case a notion of statistical parity, can help increasing the probability of exact recovery. We are able to formally explain why this phenomenon occurs, and also show empirical evidence to support our findings. Finally, as a byproduct of our analysis, we provide a tighter eigenvalue bound than that of Weyl's inequality for the case of the minimum eigenvalue.

## 2 Notation and problem formulation

Vectors and matrices are denoted by lowercase and uppercase bold faced letters respectively (e.g., $\boldsymbol{a}, \boldsymbol{A}$), while scalars are in normal font weight (e.g., $a$). For a vector $\boldsymbol{a}$, and a matrix $\boldsymbol{A}$, their entries are denoted by $a_i$ and $A_{i,j}$ respectively. Indexing starts at 1, with $\boldsymbol{A}_{i,:}$ and $\boldsymbol{A}_{:,i}$ indicating the $i$-th row and $i$-th column of $\boldsymbol{A}$ respectively. The eigenvalues of a $n \times n$ matrix $\boldsymbol{A}$ are denoted as $\lambda_i(\boldsymbol{A})$, where $\lambda_1$ and $\lambda_n$ correspond to the minimum and maximum eigenvalue respectively. Finally, the set of integers $\{1, \ldots, n\}$ is represented as $[n]$.

**Statistical parity.** In a few words, statistical (or demographic) parity enforces a predictor to be independent of the protected attributes. While the definition has been mostly used in supervised learning, in this work we try to adapt this notion of fairness to an inference problem. Specifically, we say that, given a vector attribute $\boldsymbol{a}$, the assignment $\overline{\boldsymbol{y}}$ is fair under statistical parity if $\overline{\boldsymbol{y}}^\top \boldsymbol{a} = 0$. In particular, we will consider $\overline{y}_i \in \{-1, +1\}$ to be the node labels of a graph, as described later in the problem definition. That is, we would like the partitions (or clusters) to have the same sum of the attribute $\boldsymbol{a}$.[1] As an example, we can consider the nodes of a graph to be individuals, and the node label to represent the community an individual belongs to. Then, given a vector of resources $\boldsymbol{a}$, demographic parity will enforce to output a labeling that assigns the same amount of resources to each community.

**Problem definition.** We aim to predict a vector of $n$ node labels $\widehat{\boldsymbol{y}} = (\widehat{y}_1, \ldots, \widehat{y}_n)^\top$, where $\widehat{y}_i \in \{+1, -1\}$, from a set of observations $\boldsymbol{X}$ and $\boldsymbol{c}$, where $\boldsymbol{X}$ and $\boldsymbol{c}$ correspond to noisy measurements of edges and nodes respectively. These observations are assumed to be generated from a *fair* ground truth labeling $\overline{\boldsymbol{y}}$ by a generative process defined via an undirected connected graph $G = (V, E)$, an edge noise $p \in (0, 0.5)$, and a node noise $q \in (0, 0.5)$. For each edge $(u, v) \in E$, we have a *single* independent edge observation $X_{u,v} = \overline{y}_u \overline{y}_v$ with probability $1 - p$, and $X_{u,v} = -\overline{y}_u \overline{y}_v$ with probability $p$. While for each edge $(u, v) \notin E$, the observation $X_{u,v}$ is always $0$. Similarly, for each node $u \in V$, we have an independent node observation $c_u = \overline{y}_u$ with probability $1 - q$, and $c_u = -\overline{y}_u$ with probability $q$. In addition, we are given a set of attributes $\mathbb{A} = \{\boldsymbol{a}_1, \ldots, \boldsymbol{a}_k\}$ such that $\boldsymbol{a}_i \in \mathbb{R}^n$ and $\langle \boldsymbol{a}_i, \overline{\boldsymbol{y}} \rangle = 0$ for all $i \in [k]$, i.e., for each $i$ we have $\sum_{j | \overline{y}_j = 1} (a_i)_j = \sum_{j | \overline{y}_j = -1} (a_i)_j$. In other words, we say that the ground truth labeling $\overline{\boldsymbol{y}}$ is fair under statistical parity with respect to the set of attributes $\mathbb{A}$. Thus, we have a *known* undirected connected graph $G$, an *unknown* fair ground truth label vector $\overline{\boldsymbol{y}} \in \{+1, -1\}^n$, noisy observations $\boldsymbol{X} \in \{-1, 0, +1\}^{n \times n}$ and $\boldsymbol{c} \in \{-1, +1\}^n$, a set $\mathbb{A}$ of $k$ attributes $\boldsymbol{a}_i \in \mathbb{R}^n$, and our goal is to find sufficient conditions for which we can predict, in polynomial time and with high probability, a vector label $\widehat{\boldsymbol{y}} \in \{-1, +1\}^n$ such that $\widehat{\boldsymbol{y}} = \overline{\boldsymbol{y}}$.

Given the generative process, our prediction $\widehat{\boldsymbol{y}}$ is given by the following combinatorial problem:

$$\widehat{\boldsymbol{y}} = \underset{\boldsymbol{y} \in \{-1, +1\}^n}{\arg \max} \quad \frac{1}{2} \boldsymbol{y}^\top \boldsymbol{X} \boldsymbol{y} + \alpha \cdot \boldsymbol{c}^\top \boldsymbol{y} \qquad (2)$$
$$\text{subject to} \quad \langle \boldsymbol{a}_i, \boldsymbol{y} \rangle = 0, \ \forall i \in [k]$$
$$y_i = \pm 1, \ \forall i \in [n].$$

where $\alpha = \log \frac{1-q}{q} / \log \frac{1-p}{p}$, intuitively, this value captures the amount of penalty for the linear term based on the noise parameters, and is motivated by maximum likelihood estimation (Globerson et al. 2015).

**Remark 1.** *The optimization problem 2 is clearly NP-hard to compute in general. For instance, consider the case where $k = 1$, and $(a_1)_j$ is a positive integer for all $j \in [n]$, i.e., there is a single attribute with positive entries. Also, let $\boldsymbol{X} = \boldsymbol{0}$ and $\boldsymbol{c} = \boldsymbol{0}$, that is, any vector $\boldsymbol{y}$ will attain the same objective value. Then, the problem reduces to finding an assignment $\boldsymbol{y}$ such that $\langle \boldsymbol{a}_1, \boldsymbol{y} \rangle = 0$, which is equivalent to the known NP-complete partition problem. Another example is the case when $\boldsymbol{a}_1 = \boldsymbol{1}$, that is, a feasible solution has to have the same number of positive and negative labels. Thus, if $\boldsymbol{X}$ is such that it encourages minimizing the number of edges between clusters, the problem reduces to the minimum bisection problem, which is known to be NP-complete (Garey & Johnson 1979). Finally, consider also the case in which $k = 0$, then it is known that when the graph $G$ is a grid, the problem is NP-hard (Barahona 1982).*

In the next section, we relax the combinatorial problem 2 to a continuous problem, and formally show how the addition of some fairness constraints such as that of statistical parity (as described above) can increase the exact recovery rate of previously known results (Abbe et al. 2016, Bello & Honorio 2019).

## 3 The effect of statistical parity constraint on exact recovery of labels

Our approach to analyze exact recovery will focus on the quadratic term of problem 2. This is because if $\widehat{\boldsymbol{y}} \in \{\overline{\boldsymbol{y}}, -\overline{\boldsymbol{y}}\}$ from solving only the quadratic term with the constraints, then by using majority vote with respect to the observation $\boldsymbol{c}$ one can decide which of $\{\overline{\boldsymbol{y}}, -\overline{\boldsymbol{y}}\}$ is optimal, as done by (Globerson et al. 2015, Bello & Honorio 2019). We will show sufficient conditions for exact recovery in polynomial time through the use of semidefinite programming (SDP) relaxations, which has also been previously used by (Amini & Levina 2018, Abbe et al. 2016, Bello & Honorio 2019). SDPs are optimization problems that can be solved in polynomial time by using, for example, interior point methods. Thus, showing sufficient conditions for exact recovery under SDP relaxations implies that we have sufficient conditions for exact recovery in polynomial time.

Next, we provide the SDP relaxation of problem (2). Let $\boldsymbol{Y} = \boldsymbol{y}\boldsymbol{y}^\top$, we have that $\boldsymbol{y}^\top \boldsymbol{X} \boldsymbol{y} = \mathrm{Tr}(\boldsymbol{X}\boldsymbol{Y}) = \langle \boldsymbol{X}, \boldsymbol{Y} \rangle$. Note that $\boldsymbol{y}\boldsymbol{y}^\top$ is rank-1 and symmetric, which implies that $\boldsymbol{Y}$ is a positive semidefinite matrix. Therefore, as dropping the constant $1/2$ from the quadratic term does not affect the optimal solution, the SDP relaxation to the combinatorial problem (2) results in the following

primal formulation:

$$\widehat{\boldsymbol{Y}} = \underset{\boldsymbol{Y} \in \mathbb{R}^{n \times n}}{\arg \max} \quad \langle \boldsymbol{X}, \boldsymbol{Y} \rangle \tag{3}$$

$$\text{subject to} \quad Y_{ii} = 1, \ i \in [n],$$
$$\boldsymbol{a}_i^\top \boldsymbol{Y} \boldsymbol{a}_i = 0, \ i \in [k],$$
$$\boldsymbol{Y} \succeq 0.$$

Basically, problem 3 drops the rank-1 constraint from problem 2 and results in a convex formulation that can be solved in polynomial time. Next, we present an intermediate result that is of use for the proof of Theorem 1.

**Lemma 1.** *Let $\boldsymbol{M} \in \mathbb{R}^{n \times n}$ be a positive semidefinite matrix and let $\boldsymbol{N} \in \mathbb{R}^{n \times n}$ be a rank-l positive semidefinite matrix, and consider a non-negative $\alpha \in \mathbb{R}$. Define $\Delta = \lambda_2(\boldsymbol{M}) - \lambda_1(\boldsymbol{M})$, where $\lambda_1(\cdot)$ and $\lambda_2(\cdot)$ represent the minimum and second minimum eigenvalue, respectively. Also, let $\boldsymbol{q}_1$ denote the first eigenvector of $\boldsymbol{M}$, and let $\boldsymbol{v}_1, \ldots, \boldsymbol{v}_n$ denote the eigenvectors of $\boldsymbol{N}$ related to $\lambda_1(\boldsymbol{N}), \ldots, \lambda_n(\boldsymbol{N})$ respectively. Then, we have that:*

$$\lambda_1(\boldsymbol{M} + \alpha \cdot \boldsymbol{N}) \geq \lambda_1(\boldsymbol{M}) + \max_i \left( \frac{\alpha_i + \Delta}{2} - \sqrt{\left( \frac{\alpha_i + \Delta}{2} \right)^2 - \alpha_i \cdot \Delta \cdot (\boldsymbol{v}_i^\top \boldsymbol{q}_1)^2} \right),$$

*where $\alpha_i = \alpha \cdot \lambda_i(\boldsymbol{N})$.*

*Proof.* Let $\boldsymbol{M} = \boldsymbol{Q}\boldsymbol{D}\boldsymbol{Q}^\top$ and $\boldsymbol{N} = \sum_{i=n-l+1}^{n} \lambda_i(N)\boldsymbol{v}_i\boldsymbol{v}_i^\top$ be the eigendecomposition of $\boldsymbol{M}$ and $\boldsymbol{N}$ respectively. Let us define $\boldsymbol{T} = \boldsymbol{Q}^\top(\boldsymbol{M} + \alpha \cdot \boldsymbol{N})\boldsymbol{Q}$. Since $\boldsymbol{T}$ and $(\boldsymbol{M} + \alpha \cdot \boldsymbol{N})$ are similar matrices, their spectrum is the same, which means that $\lambda_1(\boldsymbol{M} + \alpha \cdot \boldsymbol{N}) = \lambda_1(\boldsymbol{T})$. By letting $\boldsymbol{p}_i = \boldsymbol{Q}^\top \boldsymbol{v}_i$ and $\alpha_i = \alpha \cdot \lambda_i(\boldsymbol{N})$, we can express $\boldsymbol{T} = \boldsymbol{D} + \sum_{i=n-l+1}^{n} \alpha_i \cdot \boldsymbol{p}_i\boldsymbol{p}_i^\top$. Without loss of generality, consider the elements of the diagonal matrix $\boldsymbol{D}$ to be in non-decreasing order, i.e., $D_{11} = \lambda_1(\boldsymbol{M}) \leq D_{22} = \lambda_2(\boldsymbol{M}) \leq \ldots \leq D_{nn} = \lambda_n(\boldsymbol{M})$. Choose any $r \in \{n-l+1, \ldots, n\}$ and let $\widetilde{\boldsymbol{D}} = \text{diag}(D_{11}, D_{22}, \ldots, D_{22})$, and $\widetilde{\boldsymbol{T}} = \widetilde{\boldsymbol{D}} + \alpha_r \cdot \boldsymbol{p}_r\boldsymbol{p}_r^\top$. Then, we have that $\lambda_1(\boldsymbol{T}) \geq \lambda_1(\widetilde{\boldsymbol{T}})$. Denote by $\tilde{\lambda}_i$ the eigenvalues of $\widetilde{\boldsymbol{T}}$, since $\boldsymbol{p}_r\boldsymbol{p}_r^\top$ is a rank-1 matrix and $\widetilde{\boldsymbol{D}}$ has only two different eigenvalues, we have that $\tilde{\lambda}_2 = \ldots = \tilde{\lambda}_{n-1} = D_{22}$. Now,

$$\begin{aligned}
\tilde{\lambda}_1 \tilde{\lambda}_n D_{22}^{n-2} = \det(\widetilde{\boldsymbol{D}} + \alpha_r \cdot \boldsymbol{p}_r\boldsymbol{p}_r^\top) &= \det(\widetilde{\boldsymbol{D}}) \det(\boldsymbol{I} + \alpha_r \cdot \widetilde{\boldsymbol{D}}^{-1}\boldsymbol{p}_r\boldsymbol{p}_r^\top) \\
&= (1 + \alpha_r \cdot \boldsymbol{p}_r^\top \widetilde{\boldsymbol{D}}^{-1}\boldsymbol{p}_r) \det(\widetilde{\boldsymbol{D}}) \\
&= D_{11}D_{22}^{n-1}\left( 1 + \alpha_r \frac{p_{r_1}^2}{D_{11}} + \alpha_r \frac{1}{D_{22}}(1 - p_{r_1}^2) \right),
\end{aligned}$$

where the third equality comes from $\det(\boldsymbol{I} + \boldsymbol{A}\boldsymbol{B}) = \det(\boldsymbol{I} + \boldsymbol{B}\boldsymbol{A})$, and the last equality is due to $\|\boldsymbol{p}_r\|_2 = 1$. Simplifying on both ends, we obtain:

$$\tilde{\lambda}_1 \tilde{\lambda}_n = \alpha_r D_{11} + D_{11}D_{22} + \alpha_r p_{r_1}^2 \Delta \tag{4}$$

From calculating the trace we have:

$$\tilde{\lambda}_1 + (n-2)D_{22} + \tilde{\lambda}_n = \text{Tr}(\widetilde{\boldsymbol{T}}) = \text{Tr}(\widetilde{\boldsymbol{D}}) + \alpha_r \text{Tr}(\boldsymbol{p}_r\boldsymbol{p}_r^\top) = D_{11} + (n-1)D_{22} + \alpha_r.$$

Simplifying on both ends, we obtain:

$$\tilde{\lambda}_1 + \tilde{\lambda}_n = D_{11} + D_{22} + \alpha_r. \tag{5}$$

Combining eq.(4) and eq.(5), and simplifying for $\tilde{\lambda}_1$ we have, $\tilde{\lambda}_1 = D_{11} + \frac{\alpha_r + \Delta}{2} \pm \sqrt{(\frac{\alpha_r + \Delta}{2})^2 - \alpha_r \cdot \Delta \cdot p_{r_1}^2}$. Finally, since $\lambda_1(\boldsymbol{T}) \geq \lambda_1(\widetilde{\boldsymbol{T}}) = \tilde{\lambda}_1$ and the choice of $r$ was arbitrary, we take the negative sign of the square root for a lower bound and we can maximize over the choice of $r$ for the tightest lower bound. $\square$

**Remark 2.** *Note that Lemma 1 is tighter than general eigenvalue inequalities such as Weyl's inequality. Lemma 1 is tight with respect to $\Delta$ in the sense that when $\boldsymbol{N}$ is rank-1 and $\Delta = 0$, i.e., when $\lambda_1(\boldsymbol{M}) = \lambda_2(\boldsymbol{M})$, our lower bound yields $\lambda_1(\boldsymbol{M})$, which is exactly the case as the minimum eigenvalue cannot be perturbed by a rank-1 matrix under this scenario. Similarly, our bound is tight with respect to $\alpha$. When $\alpha = 0$, i.e., no perturbation, our lower bound results in $\lambda_1(\boldsymbol{M})$.*

For a graph $G = (V, E)$, its Laplacian is defined as $\boldsymbol{L}_G = \boldsymbol{D}_G - \boldsymbol{A}_G$, where $\boldsymbol{D}_G$ is a diagonal matrix with entries corresponding to the node degrees, i.e., $D_{i,i} = \deg(v_i)$ for $v_i \in V$, and $\boldsymbol{A}_G$ is the adjacency matrix of $G$. For any subset $S \subset V$, we denote its complement by $S^C$ such that $S \cup S^C = V$ and $S \cap S^C = \emptyset$. Furthermore, let $E(S, S^C) = \{(i, j) \in E \mid (i \in S, j \in S^C) \text{ or } (j \in S, i \in S^C)\}$, i.e., $|E(S, S^C)|$ denotes the number of edges between $S$ and $S^C$.

**Definition 1** (Edge Expansion). *For a set $S \subset V$ with $|S| \leq {}^n/_2$, its edge expansion, $\phi_S$, is defined as: $\phi_S = {}^{|E(S,S^C)|}/_{|S|}$. Then, the edge expansion of a graph $G = (V, E)$ is defined as: $\phi_G = \min_{S \subset V, |S| \leq {}^n/_2} \phi_S$.*

In the graph theory literature, $\phi_G$ is also known as the *Cheeger constant*, due to the geometric analogue defined by Cheeger (1969); while the second smallest eigenvalue of $\boldsymbol{L}_G$ and its respective eigenvector are known as the *algebraic connectivity* and the *Fiedler vector*[2], respectively. The following theorem corresponds to our main result where we formally show how the effect of the statistical parity constraint improves the probability of exact recovery.

**Theorem 1.** *Let $G = (V, E)$ be an undirected connected graph with $n$ nodes, Cheeger constant $\phi_G$, Fiedler vector $\boldsymbol{\pi}_2$, and maximum node degree $\deg_{\max}(G)$. Also let $\Delta$ denote the gap between the third minimum and second minimum eigenvalue of the Laplacian of $G$, namely, $\Delta = \lambda_3(\boldsymbol{L}_G) - \lambda_2(\boldsymbol{L}_G)$. Let $\boldsymbol{N} = \sum_{i=1}^{k} \boldsymbol{a}_i \boldsymbol{a}_i^\top$ with eigenvalues $\lambda_i(\boldsymbol{N})$ and related eigenvectors $\boldsymbol{v}_i$ for $i \in [n]$. Then, for the combinatorial problem (2), a solution $\boldsymbol{y} \in \{\overline{\boldsymbol{y}}, -\overline{\boldsymbol{y}}\}$ is achievable in polynomial time by solving the SDP based relaxation (3), with probability at least $1 - 2n \cdot e^{\frac{-3(\epsilon_1 + \epsilon_2)^2}{24\sigma^2 + 8R(\epsilon_1 + \epsilon_2)}}$, where*

$$\epsilon_1 = \max_{i=n-k+1\dots n} \left( \frac{n\lambda_i(\boldsymbol{N}) + \Delta}{2} - \sqrt{\left(\frac{n\lambda_i(\boldsymbol{N}) + \Delta}{2}\right)^2 - n\lambda_i(\boldsymbol{N}) \cdot \Delta \cdot (\boldsymbol{v}_i^\top \boldsymbol{\pi}_2)^2} \right),$$

$$\epsilon_2 = (1 - 2p)\frac{\phi_G^2}{4\deg_{\max}(G)}, \quad \sigma^2 = 4p(1-p)\deg_{\max}(G), \quad R = 2(1-p),$$

*and $p$ is the edge noise from our model.*

*Proof.* The dual of problem 3 is given by:

$$\min_{\boldsymbol{V}, \boldsymbol{\rho}} \quad \operatorname{Tr}(\boldsymbol{V}) \tag{6}$$

$$\text{subject to} \quad \boldsymbol{V} - \boldsymbol{X} - \sum_{i=1}^{k} \rho_i \cdot \boldsymbol{a}_i \boldsymbol{a}_i^\top \succeq 0,$$

$$\boldsymbol{V} \text{ is diagonal.}$$

Letting $\boldsymbol{\Lambda} \stackrel{\text{def}}{=} \boldsymbol{\Lambda}(\boldsymbol{V}, \boldsymbol{\rho}) = \boldsymbol{V} - \boldsymbol{X} - \sum_{i=1}^{k} \rho_i \cdot \boldsymbol{a}_i \boldsymbol{a}_i^\top$, with $\boldsymbol{V}$ diagonal. The Karush-Kuhn-Tucker (KKT) (Boyd & Vandenberghe 2004) optimality conditions are:

1. Primal Feasibility: $Y_{ii} = 1$, $\boldsymbol{a}_i^\top \boldsymbol{Y} \boldsymbol{a}_i = 0$, $\boldsymbol{Y} \succeq 0$.
2. Dual Feasibility: $\boldsymbol{\Lambda} \succeq 0$.
3. Complementary Slackness: $\langle \boldsymbol{\Lambda}, \boldsymbol{Y} \rangle = 0$.

Our approach is to find a pair of primal and dual solutions that simultaneously satisfy all KKT conditions above. Then, the pair witnesses strong duality between the primal and dual problems, which means that the pair is optimal. It is clear that $\boldsymbol{Y} = \overline{\boldsymbol{Y}} = \overline{\boldsymbol{y}}\,\overline{\boldsymbol{y}}^\top$ satisfies the primal constraints. Let $V_{ii} = (\boldsymbol{X}\overline{\boldsymbol{Y}})_{ii}$ and $\rho_i = -n$, if $\boldsymbol{\Lambda} \succeq 0$ then $\boldsymbol{V}$ and $\rho$ satisfy the dual constraints. Thus, we conclude that if the condition $\boldsymbol{\Lambda} \succeq 0$ is met then $\overline{\boldsymbol{Y}}$ is an optimal solution.

For arguing about uniqueness, let us consider that $\lambda_2(\boldsymbol{\Lambda}) > 0$ and let $\widetilde{\boldsymbol{Y}}$ be another optimal solution to problem 3. From dual feasibility and complementary slackness we have that $\boldsymbol{\Lambda}\overline{\boldsymbol{y}} = 0$, which implies that $\overline{\boldsymbol{y}}$ spans all the null space of $\boldsymbol{\Lambda}$ since $\lambda_2(\boldsymbol{\Lambda}) > 0$. Finally, from primal feasibility we have that $\widetilde{\boldsymbol{Y}} = \overline{\boldsymbol{y}}\,\overline{\boldsymbol{y}}^\top$. Thus, $\lambda_2(\boldsymbol{\Lambda}) > 0$ is a sufficient condition for uniqueness.

From the arguments above, showing the condition $\lambda_2(\boldsymbol{\Lambda}) > 0$ suffices to guarantee that $\boldsymbol{Y} = \overline{\boldsymbol{Y}}$ is optimal and unique. As $\boldsymbol{X}$ and $\boldsymbol{V}$ are random variables by construction, we next show when this condition is satisfied with high probability. By Weyl's theorem on eigenvalues, we have

$$\lambda_2(\boldsymbol{\Lambda}) = \lambda_2(\boldsymbol{\Lambda} - \mathbb{E}[\boldsymbol{\Lambda}] + \mathbb{E}[\boldsymbol{\Lambda}]) \geq \lambda_2(\mathbb{E}[\boldsymbol{\Lambda}]) + \lambda_1(\boldsymbol{\Lambda} - \mathbb{E}[\boldsymbol{\Lambda}])$$

Let $\boldsymbol{M} = \boldsymbol{V} - \boldsymbol{X}$ and $\boldsymbol{N} = \sum_{i=1}^{k} \boldsymbol{a}_i \boldsymbol{a}_i^\top$, then we have $\mathbb{E}[\boldsymbol{\Lambda}] = \mathbb{E}[\boldsymbol{M}] + n \cdot \boldsymbol{N}$, where we remove the expectation on $\boldsymbol{N}$ since it is not a random matrix. To lower bound $\lambda_2(\mathbb{E}[\boldsymbol{M}] + n \cdot \boldsymbol{N})$, we first note that $\overline{\boldsymbol{y}} \in \{\mathsf{Null}(\boldsymbol{M}) \cap \mathsf{Null}(\boldsymbol{N})\}$, which means that we can invoke Lemma 1 for $\lambda_2$ instead of $\lambda_1$. Thus, we have

$$\lambda_2(\mathbb{E}[\boldsymbol{M}] + n \cdot \boldsymbol{N}) \geq \lambda_2(\mathbb{E}[\boldsymbol{M}]) + \epsilon_1 \tag{7}$$
$$\geq \epsilon_2 + \epsilon_1, \tag{8}$$

where $\epsilon_1 = \max_{i=n-k+1...n} \left( \frac{n\lambda_i(\boldsymbol{N}) + \Delta}{2} - \sqrt{\left(\frac{n\lambda_i(\boldsymbol{N})+\Delta}{2}\right)^2 - n\lambda_i(\boldsymbol{N}) \cdot \Delta \cdot (\boldsymbol{v}_i^\top \boldsymbol{\pi}_2)^2} \right)$ in eq.(7)

follows from Lemma 1, and $\epsilon_2 = (1 - 2p) \frac{\phi_G^2}{4 \deg_{\max}(G)}$ in eq.(8) follows from Theorem 1 in (Bello & Honorio 2019). The term $\boldsymbol{\pi}_2$ in $\epsilon_1$ corresponds to the Fiedler vector of $G$ because the matrix $\boldsymbol{M}$ is a signed Laplacian of $G$ (Bello & Honorio 2019), that is, the matrix $\boldsymbol{L}_G$ and $\boldsymbol{M}$ share the same spectrum, and the $i$-th eigenvector of $\boldsymbol{M}$ is equal to the $i$-th eigenvector of $\boldsymbol{L}_G$ multiplied by $\bar{y}_i$. Since $\bar{y}_i^2 = 1$, only the second eigenvector of $\boldsymbol{L}_G$ appears in the expression, i.e., $\boldsymbol{\pi}_2$.

To lower bound $\lambda_1(\boldsymbol{\Lambda} - \mathbb{E}[\boldsymbol{\Lambda}])$, we first observe that $\boldsymbol{\Lambda} - \mathbb{E}[\boldsymbol{\Lambda}] = \boldsymbol{V} - \boldsymbol{X} - \mathbb{E}[\boldsymbol{V} - \boldsymbol{X}]$. Thus, we can further decompose the lower bound as follows: $\lambda_1(\boldsymbol{V} - \boldsymbol{X} - \mathbb{E}[\boldsymbol{V} - \boldsymbol{X}]) \geq \lambda_1(\boldsymbol{V} - \mathbb{E}[\boldsymbol{V}]) + \lambda_1(\mathbb{E}[\boldsymbol{X}] - \boldsymbol{X})$. Finally, for $\lambda_1(\boldsymbol{V} - \mathbb{E}[\boldsymbol{V}])$ and $\lambda_1(\mathbb{E}[\boldsymbol{X}] - \boldsymbol{X})$ we use Bernstein's inequality (Tropp 2012) with a similar setting to the one in the proof of Theorem 2 in (Bello & Honorio 2019) and obtain:

$$P\left(\lambda_1(\boldsymbol{V} - \mathbb{E}[\boldsymbol{V}]) \leq -\frac{\epsilon_1 + \epsilon_2}{2}\right) \leq n \cdot e^{\frac{-3(\epsilon_1+\epsilon_2)^2}{24\sigma^2 + 8R(\epsilon_1+\epsilon_2)}}, \tag{9}$$

$$P\left(\lambda_1(\mathbb{E}[\boldsymbol{X}] - \boldsymbol{X}) \leq -\frac{\epsilon_1 + \epsilon_2}{2}\right) \leq n \cdot e^{\frac{-3(\epsilon_1+\epsilon_2)^2}{24\sigma^2 + 8R(\epsilon_1+\epsilon_2)}}, \tag{10}$$

where $\sigma^2 = 4p(1-p)\deg_{\max}(G)$ and $R = 2(1-p)$. Combining equations (8), (9) and (10) we conclude our proof. $\square$

**Remark 3.** *In the proof of Theorem 1, we set $\rho_i$ to be an arbitrary finite number, in this case $\rho_i = -n$. Consider for a moment that we leave $\rho_i = \rho$ unspecified, then, in eq.(7) the goal would be to find a lower bound to $\lambda_2(\mathbb{E}[\boldsymbol{M}] - \rho\boldsymbol{N})$. Given that $\mathbb{E}[\boldsymbol{M}]$ and $\boldsymbol{N}$ are positive semidefinite matrices, then, intuitively the optimal value for $\rho$ would be $-\infty$ as it would maximize the increase in $\lambda_2(\mathbb{E}[\boldsymbol{M}] - \rho\boldsymbol{N})$. However, computationally speaking, one can note that such an assignment will never happen. Instead, any SDP solver will try to set $\rho_i$ a finite value as to observe $\lambda_2(\boldsymbol{\Lambda}) > 0$. This would be equivalent to fix $\rho$ and let the Fiedler vector $\boldsymbol{\pi}_2$ scale as to maximize $\epsilon_1$. For example, let the Fiedler vector have a norm of $\sqrt{n}$, then in this case $\epsilon_1 \to \infty$ as $n \to \infty$.*

## 4 Discussion

In this section we analyze the implications of our results through theoretical and empirical comparisons. We start by contrasting our result in Theorem 1 to previously known bounds that *did not* incorporate fairness constraints (Bello & Honorio 2019, Abbe et al. 2016). Since (Abbe et al. 2016, Bello & Honorio 2019) present bounds that are of similar rates, we take the bound from (Bello & Honorio 2019) as their bound is in a similar format than that of ours.

Following our notation, the authors in (Bello & Honorio 2019) show that the probability of error for exact recovery is $2n \cdot e^{\frac{-3\epsilon_2^2}{24\sigma^2 + 8R\epsilon_2}}$, while our result in Theorem 1 is $2n \cdot e^{\frac{-3(\epsilon_1+\epsilon_2)^2}{24\sigma^2 + 8R(\epsilon_1+\epsilon_2)}}$. Then, we can conclude that, whenever $\epsilon_1 > 0$, the *probability of error* when adding a statistical parity constraint (our model) is *strictly less* than the case with no fairness constraint whatsoever (models studied in (Abbe et al. 2016, Bello & Honorio 2019, Globerson et al. 2015, Foster et al. 2018)).

The above argument poses the question on when $\epsilon_1 > 0$. Recall from Theorem 1 that $\epsilon_1 = \max_{i=n-k+1...n} \left( \frac{n\lambda_i(\mathbf{N}) + \Delta}{2} - \sqrt{\left( \frac{n\lambda_i(\mathbf{N}) + \Delta}{2} \right)^2 - n\lambda_i(\mathbf{N}) \cdot \Delta \cdot (\mathbf{v}_i^\top \boldsymbol{\pi}_2)^2} \right)$. For clarity purposes, we discuss the case of a single fairness constraint, that is, $\mathbf{N} = \mathbf{a}_1 \mathbf{a}_1^\top$, and let $\|\mathbf{a}_1\|_2^2 = s$. Then we have that $\epsilon_1 = \frac{n \cdot s + \Delta}{2} - \sqrt{\left( \frac{n \cdot s + \Delta}{2} \right)^2 - n \cdot \Delta \cdot (\mathbf{a}_1^\top \boldsymbol{\pi}_2)^2}$, from this expression, it is clear that whenever $\Delta > 0$ and $\langle \mathbf{a}_1, \boldsymbol{\pi}_2 \rangle \neq 0$ then $\epsilon_1 > 0$. In other words, to observe improvement in the probability of exact recovery, it suffices to have a non-zero scalar projection of the attribute $\mathbf{a}_1$ onto the Fiedler vector $\boldsymbol{\pi}_2$, and an algebraic connectivity of multiplicity 1.[3] Finally, note that since $\langle \mathbf{a}_1, \boldsymbol{\pi}_2 \rangle$ depends on $\mathbf{a}_1$, which is a given attribute, one can safely assume that $\langle \mathbf{a}_1, \boldsymbol{\pi}_2 \rangle \neq 0$. However, the eigenvalue gap $\Delta$ depends solely on the graph $G$ and raises the question on what classes of graphs we observe (or do not) $\Delta = 0$.

## 4.1 On the multiplicity of the algebraic connectivity

Since $\Delta > 0$ if and only if the multiplicity of the algebraic connectivity is 1, we devote this section to discuss in which cases this condition does or does not occur. After the seminal work of Fiedler (1973), which unveiled relationships between graph properties and the second minimum eigenvalue of the Laplacian matrix, several researchers aimed to find additional connections. In the graph theory literature, one can find analyses on the complete spectrum of the Laplacian (e.g. (Grone et al. 1990, Grone & Merris 1994, Mohar et al. 1991, Newman 2001, Das 2004)), where the main focus is to find bounds for the Laplacian eigenvalues based on structural properties of the graph. Another line of work studies the changes on the Laplacian eigenvalues after adding or removing edges in $G$ (Kirkland 2005, 2010, Barik & Pati 2005). To our knowledge the only work who attempts to characterize families of graphs that have algebraic connectivity with certain multiplicity is the work of (Barik & Pati 2005). Let $\boldsymbol{\pi}$ be a Fiedler vector of $G$, we denote the entry of $\boldsymbol{\pi}$ corresponding to vertex $u$ as $\pi_u$. A vertex $u$ is called a *characteristic vertex* of $G$ if $\pi_u = 0$ and if there exists a vertex $w$ adjacent to $u$ such that $\pi_w \neq 0$. An edge $\{u, w\}$ is called a *characteristic edge* of $G$ if $\pi_u \pi_w < 0$. The *characteristic set* of $G$ is denoted by $C_G(\boldsymbol{\pi})$ and consists of all the characteristic vertices and characteristic edges of $G$. Let $W$ be any proper subset of the vertex set of $G$, by a branch at $W$ of $G$ we mean a component of $G \setminus W$. A branch at $W$ is called a *Perron branch* if the principal submatrix of $\mathbf{L}_G$, corresponding to the branch, has an eigenvalue less than or equal to $\lambda_2(\mathbf{L}_G)$. The following was presented in (Barik & Pati 2005) and characterizes graphs that have algebraic connectivity with certain multiplicity.

**Theorem 2** (Theorem 10 in (Barik & Pati 2005)). *Let $G$ be a connected graph and $\boldsymbol{\pi}$ be a Fiedler vector with $W = C_G(\boldsymbol{\pi})$ consisting of vertices only. Suppose that there are $t \geq 2$ Perron branches $G_1, \ldots, G_t$ of $G$ at $W$. Then the following are equivalent.*

- *The multiplicity of $\lambda_2(\mathbf{L}_G)$ is exactly $t - 1$.*
- *For each Fiedler vector $\boldsymbol{\psi}$, $C_G(\boldsymbol{\psi}) = W$.*
- *For each Fiedler vector $\boldsymbol{\psi}$, the set $C_G(\boldsymbol{\psi})$ consists of vertices only.*

The above characterization is very limited in the sense that authors in (Barik & Pati 2005) are able to show only one example of graph family that satisfies the conditions above. Specifically, their example correspond to the class $G = (K_{n-t}^C + H_t^C)^C$, where $K_i$ denotes the complete graph of order $i$ and $H_j$ is a graph of $j$ isolated vertices, and for $G_1 = (V_1, E_1), G_2 = (V_2, E_2)$, the operation $G = G_1 + G_2$ is defined as $G = (V_1 \cup V_2, E_1 \cup E_2)$. A particularly known instance of this class is $t = n - 1$, which corresponds to the star graph and has algebraic connectivity with multiplicity $n - 2$ and therefore $\Delta = 0$ for $n > 3$.

Another known example where $\Delta = 0$ is the complete graph $K_n$ of order $n$ where there is only one non-zero eigenvalue equal to $n$ and with multiplicity $n - 1$. We now turn our attention to graphs with poor expansion properties such as grids. A $m \times n$ grid, denoted by $\text{Grid}(m, n)$, is a connected graph such that it has 4 corner vertices which have two edges each, $m - 2$ vertices that have 3 edges which make up the short "edge of a rectangle" and $n - 2$ vertices that have 3 edges each which make up the

"long edge of a rectangle" and $(n-2)(m-2)$ inner vertices which each have four edges. (Edwards 2013) characterizes the full Laplacian spectrum for grid graphs as follows: the eigenvalues of the Laplacian matrix of $\mathrm{Grid}(m,n)$ are of the form $\lambda_{i,j} = (2\sin(\frac{\pi i}{2n}))^2 + (2\sin(\frac{\pi j}{2m}))^2$, where $i$ and $j$ are non-negative integers. Next, we present a corollary showing the behavior of $\Delta$ in grids.

**Corollary 1.** *Let $G$ be a grid graph,* $\mathrm{Grid}(m,n)$*, then we have:*

- *If $m = n$ then $\Delta = 0$.*
- *If $m \neq n$ then $\Delta > 0$.*

*Proof.* Since $\lambda_{i,j} = (2\sin(\frac{\pi i}{2n}))^2 + (2\sin(\frac{\pi j}{2m}))^2$, then $\lambda_{i,j} = 0$ if and only if $(i,j) = (0,0)$ and corresponds to the first eigenvalue of the Laplacian. It is clear that the next minimum should be of the form $\lambda_{0,j}$ and $\lambda_{i,0}$. By taking derivatives we obtain: $\frac{d\lambda_{i,0}}{di} = \frac{2\pi}{n}\sin(\frac{\pi i}{n})$ and $\frac{d\lambda_{0,j}}{dj} = \frac{2\pi}{m}\sin(\frac{\pi j}{m})$. We observe that the minimums are attained at $\lambda_{1,0} = (2\sin(\frac{\pi}{2n}))^2$ and $\lambda_{0,1} = (2\sin(\frac{\pi}{2m}))^2$ respectively. Thus, when $m = n$ we have $\Delta = 0$ and when $m \neq n$ we have $\Delta > 0$. ☐

That is, Corollary 1 states that square grids have $\Delta = 0$, while rectangular grids have $\Delta > 0$. To conclude our discussion on $\Delta$, we empirically show that the family of Erdős-Rényi graphs exhibit $\Delta > 0$ with high probability. Specifically, we let $G \sim \mathsf{ER}(n,r)$, where $r$ is the edge probability. When $r = 1$, $G$ is the complete graph of order $n$ and $\Delta > 0$ with probability zero. Interestingly, when $r = 0.9$ or $r = 0.99$, that is, values close to 1, the probability of $\Delta > 0$ tends to 1 as $n$ increases. Also, we analyze the case when $r = 2\log n/n$,[4] and also observe high probability of $\Delta > 0$. The aforementioned results are depicted in Figure 1 (Left). Intuitively, this suggests that the family of graphs where $\Delta > 0$ is much larger than the families where $\Delta = 0$. Finally, in Figure 1 (Right), we also plot the expected value of the gap, where we note an interesting concentration of the gap to $0.5$ for $r = 2\log n/n$. An explanation of the latter gap behavior remains an open question.

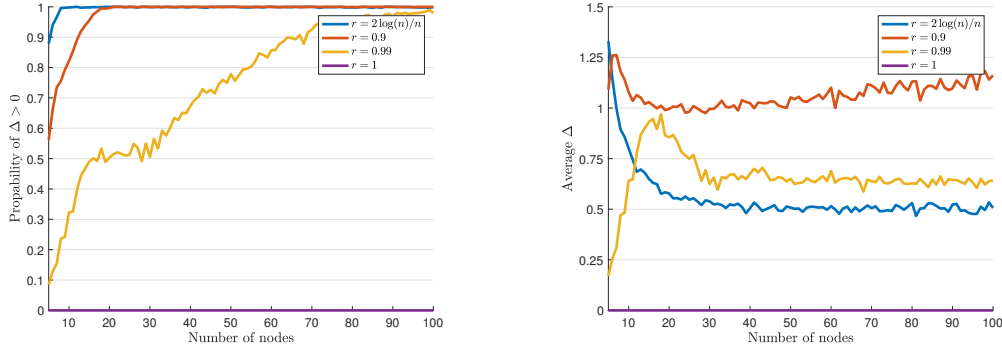

Figure 1: Graphs drawn from an Erdős-Rényi model with $n$ nodes and edge probability $r$. *(Left.)* Probability of $\Delta > 0$ for each number of nodes, we draw 1000 graphs and compute $\Delta$, then, we count an event as success whenever $\Delta > 0$, and failure when $\Delta = 0$. *(Right.)* Expected value of $\Delta$ computed across the 1000 random graphs for each number of nodes.

### 4.2 Experiments

In this section, we corroborate our theoretical results through synthetic experiments. Graphs with high expansion properties such as complete graphs and $d$-regular expanders are known to manifest high probability of exact recovery as their Cheeger constant increases with respect to $n$ or $d$ (Bello & Honorio 2019). That is, in these graphs, the effect of the fairness constraint will not be noticeable. In contrast, graphs with poor expansion properties such as grids, which have a Cheeger constant in the order of $\mathcal{O}(1/n)$ for a $\mathrm{Grid}(n,n)$, can only be recovered approximately (Globerson et al. 2015), or exactly if the graph can be perturbed with additional edges (Bello & Honorio 2019). Thus, we focus our experiments on grids and empirically show how the inclusion of the fairness constraint boosts the

probability of exact recovery. In Figure 2, we first randomly set $\overline{\boldsymbol{y}}$ by independently sampling each $\overline{y}_i$ from a Rademacher distribution. We consider a graph of $64$ nodes, specifically, $\mathrm{Grid}(4, 16)$, i.e., $\Delta$ is guaranteed to be greater than 0. Finally, we compute 30 observations for $p \in [0, 0.1]$. When there is no fairness constraint, we observe that the probability of exact recovery decreases at a very high rate, while the addition of fairness constraints improves the exact recovery probability. In particular, we note that while the addition of a *single* fairness constraint (SDP + 1F) helps to achieve exact recovery, the tendency is to still decrease as $p$ increases, in this case the attribute $\boldsymbol{a}_1$ was randomly sampled from the nullspace of $\overline{\boldsymbol{y}}^\top$ so that $\overline{\boldsymbol{y}}^\top \boldsymbol{a}_1 = 0$. We also show the case when two fairness constraints are added (SDP + 2F), were we observe that exact recovery happens almost surely, here the two attributes also come randomly from the nullspace of $\overline{\boldsymbol{y}}^\top$.

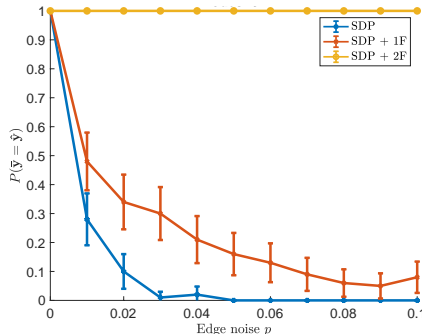

Figure 2: Probability of exact recovery for $\mathrm{Grid}(4, 16)$ computed across 30 observations $\boldsymbol{X}$ for different values of $p \in [0, 0.1]$. We observe how the addition of fairness constraints helps exact recovery, where SDP+1F refers to the addition of a single constraint, and SDP+2F the addition of two constraints.

## 5    Concluding remarks

We considered a model similar to that of (Globerson et al. 2015, Bello & Honorio 2019, Foster et al. 2018, Abbe et al. 2016) and studied the effect of adding fairness constraints, specifically, under a notion of statistical parity, and showed how it can help increase the probability of exact recovery even for graphs with poor expansion properties such as grids. Note that, given that our definition of statistical parity is a linear constraint, our analysis and results will hold for any linear constraint not necessarily attached to a fairness viewpoint.

In our analysis, we assumed that the ground-truth labeling is fair. While the linear constraints reduce the search space in the relaxed continuous problem, before our results, it was unclear how these constraints affect the probability of exact recovery, which we formally show in Theorem 1. We argue that even in the scenario of having "fair data" one should not rule out the possibility of adding fairness constraints as there is a chance that it can help increase the performance. For instance, a practitioner could use one of the several *preprocessing* methods for debiasing a dataset with respect to a particular metric (Zemel et al. 2013, Calmon et al. 2017, Louizos et al. 2015, Gordaliza et al. 2019), assuming that the data is now fair, the practitioner might be tempted not to use any fairness constraint anymore. However, as showed in this work, when the data is fair, adding fairness constraint can improve performance. As future work, it might be interesting to analyze different soft versions of the generative model such as allowing the data being at most $\varepsilon$-away with respect to some fairness criteria instead of imposing a hard constraint.

## Broader Impact

This theoretical work does not present any foreseeable societal consequence. We believe the research community interested in the foundations of machine learning might benefit from the results of this work.

## Acknowledgments and Disclosure of Funding

This material is based upon work supported by the National Science Foundation under Grant No. 1716609-IIS.

## Footnotes

[1] Note that the elements of the attribute can already be divided by the size of the clusters they belong to, in which case it would represent equal averages. Here we make no assumptions on the elements of $\boldsymbol{a}$.

[2]If the multiplicity of the algebraic connectivity is greater than one then we have a set of Fiedler vectors.

[3]Specifically, we refer to the algebraic multiplicity. Having an algebraic connectivity with multiplicity greater than 1 will imply that $\Delta = 0$.

[4]Our motivation for the choice of $r = 2\log n/n$ is that for $r > (1+\varepsilon)\log n/n$ then the graph is connected almost surely (Erdős & Rényi 1960).

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
