[Reviews · NeurIPS 2020]

Review 1

Summary and Contributions: This paper analyzes local inference in structured prediction with an additional 'fairness' constraint. They show that this constraint improves the probability of recovering the ground-truth labels over previous approaches without this constraint via a SDP relaxation. They also demonstrate that exact recovery in a toy grid graph is much more likely when adding in a randomly chosen constraint.

Strengths: This is relevant, very interesting work with non-trivial proofs. I do not know of any other results that take advantage of this constraint in this manner. While I am not an expert in SDP relaxations, it is truly surprising to me that the additional constraint helps. Edit: Thanks for the response.

Weaknesses: The experiments section could have used a few more examples. It's not clear to me if these results should hold for other grids with m\neq n: why Grid(4,16)? Or more realistic graphs for which we would actually want to do inference, which will not be grids or d-regular graphs. I also would appreciate more intuition on how the additional constraint really helps. The proofs, at least to me, appear to hide the intuition more than explain it.

Correctness: I did not check the proofs in detail, but they appear reasonable.

Clarity: The paper is clear and well-written; no complaints.

Relation to Prior Work: I know of no other prior work relevant to this.

Reproducibility: Yes

Additional Feedback: This paper's Broader Impacts section deserves to be taken seriously. What kind of structured prediction problems in practice would this approach benefit from? Are there any potential downsides to this approach? For example, what if a practitioner wants to do structured prediction with a fairness constraint other than statistical parity, might this be an issue?


Review 2

Summary and Contributions: This paper demonstrates that certain constraints that the authors call “fair constraints” can help to reduce the probability of error in a clustering problem [22]. The technical contribution is manifested in Lemma 1, which is also emphasized as a side byproduct in light of Weyl’s inequality.

Strengths: Development of Lemma 1.

Weaknesses: Meaning of the considered fair constraints: This reviewer wonders why the introduced constraints can be interpreted as fair constraints w.r.t. statistical parity. This interpretation is important. Otherwise, it can serve rather as “side information” for the considered setting where the ground truth labeling is assumed to be fair. Under the labeling assumption, the claim looks straightforward. It may not be the case if the ground truth labeling is inconsistent with the constraint. Hence, providing reasonable justifications behind why the constraints are fair and the labeling assumption are crucial.

Correctness: The proofs of the main result (Theorem 1) and the technical contribution (Lemma 1) look correct.

Clarity: Overall it is difficult to follow. Organizations and descriptions of the proofs can be improved.

Relation to Prior Work: The problem formulation is similar to that of [22], except for the introduced constraint. Also, many technical details bear similarity to those of [8], except Lemma 1.

Reproducibility: Yes

Additional Feedback: ============== After rebuttal ============== I have read the author response as well as other reviewrs' comments and discussions. While my concern re. fairness interpretation is addressed to some extent, I am still concerned about the "fair" labeling assumption. Specifically in experiments re. Fig. 2, \bar{y} and a1 are chosen such that their inner product is zero. This means that the constraint indeed serves as side information. So I am still wondering what if the ground truth labeling is unfair, i.e., \bar{y}^Ta_1 \neq 0. Nonetheless, I agree that adding such constraints does not make the problem trivial and the contribution of this paper is an explicit quantification of the gain due to constraints.


Review 3

Summary and Contributions: Like [8], this paper tackles the problem of recovering binary labels for the vertex of a graph given a noisy observation of the following form: for each vertex, we observe the correct label with probability 1-q, and for each edge, we observe the product of the correct labels with probability 1-p. In this paper, the following "fairness" assumption is added: some attribute vectors a_1,\ldots,a_k are given such that the ground truth labels are orthogonal to all attributes, and is known to be so by the algorithm user. Like in [8], the algorithm only uses the vertex observations to decide the sign of the final solution, with the main body of the algorithm consisting in solving a convex relaxation of the maximum likelihood function corresponding to the edge observations alone. The relaxation is a semidefinite programming problem which can be tackled with the corresponding duality theory. Like in [8], the idea is to show that with high probability, the ground truth, together with some explicitly chosen slack variables, consitute a pair that is both dual and primal feasible, and in fact is the only solution. The proof strategy involves bounding the second smallest eigenvalue of the matrix $\Lambda$ from the dual conditions. Via the Weyl inequalities, the problem is divided into estimation of the corresponding eigenvalue of the expectation of $\Lambda$ and the smallest eigenvalues of the deviation from the expectation. The latter term can be tackled as in [8] via the matrix version of the Bernstein inequality. The first term is tackled with Lemma 1, which cleverly exploits the PSD properties to lower bound the increase in the smallest eigenvalue of a psd matrix when another psd matrix is added to it. The quantity corresponding to this second term comes from Lagrangian term corresponding to the fairness conditions. The final results show that the restriction in the solution space corresponding to the fairness conditions strictly increases the probability of exact recovery.

Strengths: The results and proof techniques are sound (I went through most of the proofs once and didn't see any problems except the minor confusing points I mention below). The mathematical writing is precise (albeit rather crisp). I cannot vouch for the originality given my grasp of the literature, but the proof techniques seem reasonably interesting (especially Lemma 1). Of course, the fact that the authors are now able to show high probability recovery for the case of (strictly) rectangular grids (without the awkward assumption of a small number of extra random edges from [8]) is a huge strength as well. I like that the authors simply used the relevant results from [8], making the present proofs distinct from those in [8], better allowing us to judge the novelty of the present paper. I like that they included experiments including conjectures about asymptotic behaviour of the quantity $\Delta$. I (moderately) enjoyed the morale of the story in the conclusion, exhorting users to make use of the assumption of fairness even if the data has been rendered fair through some preprocessing step. Overall, it certainly is a strong paper paper.

Weaknesses: In decreasing order of importance: 1. My biggest concern is with the way that $\epsilon_1$ behaves depending on n. Firstly, it seems the choice of the value $-n$ for $\rho$ is arbitrary (with the choice being repercuted in the definition of $\epsilon$), and this should be discussed more clearly in the text. Next, it is not clear to me why the choice $\rho=-n$ is the best. Does it optimize $\epsilon_1$ in some way? Furthermore, as n tends to $\infty$, it seems that $\epsilon_1$ does NOT tend to infinity. This would imply that the bounds in the cases where no good bound on $\epsilon_2$ is available (i.e., the cases which are not covered by the work [8]), the final bound, although not trivial, is not a "high probability" bound in the strict sense of a vanishingly small failure probability. The closest thing to an explanation for this seems to be in the beginning of Section 4, but I could not find the answer to my question there. I assume I misunderstood something here, and I might lower my score in the less likely case that this turns out to be as serious an issue as it superficially appears to me. 2. The idea that fairness improves the bounds is not counter intuitive as claimed in the introduction: it is clear that the fairness assumption which applies to both the ground truth and the function search space reduces the complexity of the problem. (And without a good answer to point one, the improvement would be incremental from the theoretical point of view, although I agree that the experiments section shows improved results in this case). Furthermore, there is no attempt at extending the results to a slightly different setting or applying the results to any well defined machine learning problem, which undermines the relevance of the work to the community. It would also be intresting to try to tackle the case where the ground truth is not fair but the method requires fairness (this is closer to the general paradigm studied in the fairness literature). One could for instance consider the case where the ground is close to satisfying fairness (but does not exactly), as a result of being drawn from a high dimensional distribution whose expectation satisfies the fairness assumption: for instance, suppose the vertices of the graph are people and the attribute is "male/female", with the labels representing failure or success. For a finite graph, it is reasonable to assume that the ground truth labels are drawn from a distribution with each label being independent of gender. This will not necessarily translate to an exactly fair set of ground truth labels such as the ones considered in this work. It is a little underwhelming that such natural situations are not treatable with the results provided in this work. (2. bis. Still no solution for square grids...) And less importantly: 3. Although many of the proofs are impressive and rigorous, I don't feel they are very reader friendly (though it could be explained by a lack of familiarity with the literature on my part). In the Review Section "Additional feedback", I list some aspects which could be explained in greater detail. 4. There are some other very minor issues I list in the Section "Additional feedback".

Correctness: To the best of my knowledge, yes.

Clarity: Generally, yes. However, I find that starting from line 270, the writing gets worse, though it is still not too bad. There are issues with the reader friendliness, but this is less about the writing and more about the technical exposition.

Relation to Prior Work: I think so, though some of it is discussed in Section 4 instead of the related works Section.

Reproducibility: Yes

Additional Feedback: First, here are some things that made reading difficult for me. (listed in order of appearance rather than importance) 1. In the introduction, it is not explained clearly what kind of machine learning problem the specific problem studied could correspond to. 2. Throughout the whole of the main Section 3, there is one important thing to understand that is not clearly explained (not in [8] either): when we decide to drop the linear term (and when we decide to relax the rank one constraint on Y) we are not calculating a well principled approximation to the original optimization problem, but rather, throwing away information because we are confident that the information left is enough to recover the ground truth: it turns out that for the cases studied, with high probability, the combination of binary labels that best explain the observed edges is the ground truth, so it is not necessary to take the vertex labels into account in the algorithm. 3. In the description of the dual problem on line 169, there is only one variable $\rho$ for all of the fairness constraints. I understand this comes from the positive semi definiteness of $Y$, which implies that the only way that $\sum a_i^\top Y a_i=0$ could happen is if each term in the sum is zero. If this understanding is correct, it should be explained in one extra line. 4. The logical organization of lines175 to 170 feel awkward to someone not proficient in the optimization theorems used: I think complementary slackness follows from the dual and primal feasability, and it is easy to check it from scratch as well. However, in the text, the authors appear to list the three KKT conditions (including complementary slackness) as something to be proved, then prove only primal and dual feasability, and then *use* Complementary slackness later. As mentioned elsewhere in the review, the definitions of $V$ and $\rho$ are a little brusque, especially since they don't matter for the conclusion which is drawn in the same sentence (although they matter later, in the results from [8] used in equation (8)). 5. The discussion of Theorem 2 seems opaque and difficult to place in the context of the discussion (it feels like related work). Next, I list some very minor (non conceptual) issues with the mathematical formating. 1. I don't see a reason why there is a factor of 4 (resp. 2) in the definition of $\sigma^2$ if the only time the quantity is used is in a formula where there is an extra factor of 24 (resp. 8). 2. In the proof of Lemma 1, it would probably be better to write $(p_r)_1$ instead of $p_{r_1}$ to denote the first component of the vector $p_r$. 3. I would personally prefer a "\left \right" around the brackets that enclose $\frac{\alpha_i+\Delta}{2}$ in various places. Finally, here is a list of minor typos/questions which could be considered in the camera ready version: 1. Line 16: "capable to achieve" --> "capable of achieving". 2. Line 18: could be replaced by " we provide a tighter... bound than that which can be derived from Weyl's inequality. 3. Line 37 "incompatible to be achieved", one could remove the last three words. The sentence in question is also too long. 4. Line 46 could become "We study a generative model similar to the one that has been previously studied in [22,19,8,1]. 5.Line 95: "the problem reduces to find" --> "the problem reduces to finding". 6 The structure of the sentence on lines 105 to 107 could be improved (I also mentioned above issues I have with its factual clarity). 7. Line 273: "then" could be replaced by "and" 8. The statement about the open question on line 279 should be a separate sentence. 9. Footnote 4 on page 7 could be "Our motivation for the choice of $r=2\log(n)/2$ is that for $r> (1+\epsilon)\log(n)/n$, the graph is almost surely connected (cf. ~\cite{https://web.stanford.edu/class/msande235/erdos-renyi.pdf})" (for instance) 10. Lines 291-292: " decreases with a very high rate" -->"decreases at a very high rate" 11. The first sentence of the concluding remarks on page 298 is a little long/slightly awkward. 12. Line 302: "help increasing" -->" help increase" 13. Line 305 "tempted to not use " --> "tempted not to use" 14. Lines 307-308: "....such as letting the data being at most...." --> " ....such as allowing the data to be at most...." ============================================================== Post rebuttal. After reading the other reviewers' comments and the rebuttal, I have slightly increased my score. On "fairness", I still think that whilst the use of the term "fair" is slightly overselling it, the problem considered here is far from trivial and far from being a simple extension of [8]. About $\rho$, I understand it can be set to an arbitrary negative number, the authors' answers make sense (it would be nice to write more details in the paper). However, I don't feel I have received a complete answer to my question about the behavior of \epsilon_1. It is still unclear to me whether or not there exists a sequence of graphs such that \epsilon_1 \tendsto \infty but \epxilon_2 is bounded. The authors seem to say there is, but I still think if it was easy to describe it in detail they would have done so in the paper. Discussions with the other reviewers about this point were inconclusive. However, I am not an expert in the field, and I really don't want to be responsible for rejecting a paper if the reason is a misunderstanding on my part, so I am willing to give the authors the benefit of the doubt (anyway, I think the paper would still be (marginally) above the acceptance threshold if this issue were not solvable). If possible, I strongly recommend a thorough discussion of this point beyond (the short hints given in the rebuttal limited by page length) be added to the paper or the supplementary.


Review 4

Summary and Contributions: Post-rebuttal The responses and the reviewer discussion answered a lot of my questions. I increased my score. ------------------------------------ The authors study a particular problem in signal recovery. Specifically, the nodes of a graph have a binary value. We do not observe these values, but we do observe each node's value passed through a binary symmetric channel. In addition, we also observe the products of the values of neighboring nodes (these edge values also passed through the same channel). Finally, there is an additional requirement here---a known attribute vector must be orthogonal to the signal to be recovered. Solving this problem is computationally hard in general. However, a convex relaxation enables a polynomial-time computation with guarantees on recovery in certain cases, ie, without the 'fairness' constraint. The authors derive a result for the new case, based on a lemma that lower bounds the eigenvalues of the sum of two matrices, which is better than the conventional Weyl eigenvalue inequalities. Finally, the authors do experiments on synthetic graphs to validate the main result.

Strengths: - This is an interesting statistical problem, and the authors make a nice step to tackle a constrained version of it. - The result in Lemma 1 is actually interesting from the theoretical point of view. - The graph property discussion on the algebraic property of graph structures was quite good also.

Weaknesses: - The motivation of this version of the problem as being under "fairness constraints" rather than one very limited notion of fairness seems like a bit of a stretch. - The results in the paper are very close to the non-fairness version. - It's hard to say how much of a fit this is for the conference, especially as a purely theoretical signal recovery problem and no experiments on real-world data.

Correctness: In general, yes.

Clarity: For the most part, I felt things were clear, beyond a few examples in the comments below.

Relation to Prior Work: The most important work to be cited here, the Bello and Honorio '19 paper that the authors build on, is cited and discussed. The rest of the prior work seemed fine.

Reproducibility: Yes

Additional Feedback: I was lukewarm on the paper, and I voted weak accept. I think the underlying problem is interesting. The authors definitely take a step forward, and Lemma 1 is nice. However, the overall contribution and the applicability seem limited. A few additional questions: - In the interpretation of Theorem 1, the authors say that we can conclude that "whenever \epsilon_1 > 0 , the probability of error when adding a statistical parity constraint (our model) is strictly less than the case with no fairness constraint whatsoever". What am I missing here? Sees like the result could be smaller or larger depending on \epsilon_1 and the other terms. - It'd be interesting to discuss what other types of constraints could be considered (clearly, other linear constraints could also work), as that could encompass more notions of fairness.

[Author Response · NeurIPS 2020]

We thank all reviewers for their comments and feedback. In what follows we try to address the main concerns, we encourage
reviewers to read all our responses as some are related.

**R1:** We thank R1 for appreciating our results and efforts put in our work. • Our theory does not make any assump-
tion on the underlying graph except that it is connected. Thus, Theorem 1 holds for grids of different sizes or graphs
other than grids. In the next figure, we sampled an Erdős-Rényi graph (used in analysis of social networks, e.g., "Ran-
dom graph models of social networks" by Newman et al. (2002)) of 100 nodes with edge probability of 0.1382 ($3\log n/n$).
The rest of the setting follows that of Section 4.2.
As discussed in Section 4.1, we have that $\Delta > 0$

with high probability, and we can observe in the
left plot how the addition of a single constraint
boosts the probability of exact recovery. The right
plot shows the sampled graph with red and blue
nodes indicating the true $-1$ and $+1$ labels re-
spectively. • One intuitive way to understand the
improvement is by looking at eq.(7) in page 5.
Without the constraint, the problem is equivalent
to that of [8]. However, when the linear constraint
is added, in the dual problem of the SDP relax-
ation, it appears a new term $N$ that is a PSD
matrix, hence, it cannot lower the value of $\lambda_2(\mathbb{E}[M])$. We will add a line in the manuscript explaining this.

**R2:** • The only part we borrowed from [8] is the lower bound on $\lambda_2(\mathbb{E}[M])$ and the setting of the dual variable $V$ from [1] and
[8]. The remaining part of our work (Lemma 1, adaption of Lemma 1 into Theorem 1, Corollary 1, Discussion on connections to
eigenvalue gap in the Laplacian matrix and Fiedler vector, and Empirical evidence) is novel as noted by R1, R3. • We disagree that
the overall content is difficult to follow. R1, R3 and R4 stated that the presentation was mostly clear. • Regarding the meaning of the
constraint, let us provide an example how it can be interpreted as imposing demographic parity at *inference* time. Let the nodes in the
graph represent individuals, where the label indicates the community a person belongs to. Then, let $a$ be a vector of some resources
that ideally should be split equally to both communities. The constraint can be interpreted as forcing a labeling to create two
communities where the sum of resources is equal for each community. Finally, we also argue that even if the constraint is seen as side
information, it does **not** imply that the combinatorial problem is easier, in fact, as discussed in Remark 1, it is still NP-hard in general.
We can empirically corroborate that in some sce-

narios the addition of a **single** constraint does not
improve the probability of exact recovery. In the
next figure, we follow a same setting to that of
Section 4.2, where now the graphs are grids of
6x6 (left) and 16x16 (right). We observe how
the addition of a single constraint does not help
exact recovery as suggested by our discussion on
$\Delta = 0$ for square grids, however, the addition
of two constraints can help square grids to be
recovered exactly.

**R3:** • Regarding the major concern, we confirm that $\rho$ was set to be an arbitrary finite number, in this case $-n$. Consider for a
moment that we leave $\rho$ unset, then in eq.(7) the goal would be to find a lower bound to $\lambda_2(\mathbb{E}[M] - \rho N)$. Given that $\mathbb{E}[M]$ and $N$
are PSD matrices, then intuitively the optimal setting for $\rho$ would be $-\infty$ as it would maximize the increase in $\lambda_2(\mathbb{E}[M] - \rho N)$.
However, computationally speaking, one can note that such assignment will never happen. Instead, the SDP solver will try to set $\rho$ a
**finite** value as low as possible as to observe $\lambda_2(\Lambda) > 0$. This would be equivalent to fix $\rho$ and let the Fiedler vector $\pi_2$ scale as to
maximize $\epsilon_1$. For example, let the Fiedler vector have a norm of $\sqrt{n}$, then in such case $\epsilon_1$ will tend to $\infty$ as $n$ goes to $\infty$. This short
discussion will be added to the manuscript for further clarity. • We remark that even if the ground truth is fair, it does not imply
that the outcome will be fair as it is known in the fairness literature that the choice of model or algorithm has an effect in the final
outcome. Thus, while the "relaxed" setting proposed by R3 is interesting and appealing as future work, we believe our work is a
first step in the line of considering fairness constraints even in the scenario of having fair data. • Note that while square grids have
$\Delta = 0$, this only implies that a single constraint is not sufficient to observe improvement in exact recovery. However, given that the
multiplicity of the algebraic connectivity in square grids is 2, two constraints can help exact recovery, as shown in the figures above
(please see bullet 3 for R2). • We thank R3 for his thorough review and feedback. We plan to incorporate the small suggestions to
improve the paper presentation. Finally, regarding the KKT conditions, complementary slackness is an optimality condition that is
fulfilled by $Y$ and hence used to derive the sufficient condition on uniqueness.

**R4:** • As noted by R4, the constraint can have interpretations other than fairness and would be interesting, as future work, to study
other fairness notions or interpretations. However, our results would still apply whenever one has linear equality constraints. •
We disagree that the results are very close to that of non-fairness version. R1 and R3 note that we provide important connections
between linear constraints and the eigenvalue gap of the Laplacian ($\Delta$) along with the Fiedler vector $\pi_2$, which to the best of our
knowledge was unknown before. • Pure theoretical work has historically been welcomed to NeurIPS, which can be noted in the
conference website. In addition, our work was submitted to the Statistical Learning Theory category as our main contribution is in
the understanding of the effect of fairness constraints at inference time.

[Meta-Review · NeurIPS 2020]

This paper is about structures output predictions analysis under 'fairness' constraint. This paper shows that constraints relative to fairness can help to increases accuracies. Fairness is one of the notion whose importance is rising in our community, and this paper give interesting insights about it. One of the main issue raised by one of the reviewer that pleads for non acceptance is the "vagueness" of the definition of fairness here. I personally think that this issue should not be taken to much into account here, there is still in our community some "vagueness" according to what the good definition should be. Another point that has been raised by one of the reviewers is "The idea that fairness improves the bounds is not counter intuitive AS CLAIMED IN THE INTRODUCTION: it is clear that the fairness assumption which applies to both the ground truth and the function search space reduces the complexity of the problem. I agree with the reviewer that adding constraint reduces complexity, but it is nevertheless an important result to see how it can be captured by a bound. So overall, a nice paper on an important ML issue, the fairness ... clear acceptance.